# Sleep deprivation suppresses aggression in *Drosophila*

**Matthew S Kayser**[1,2,4]*, **Benjamin Mainwaring**[1], **Zhifeng Yue**[3], **Amita Sehgal**[2,3,4]*

[1]Department of Psychiatry, Perelman School of Medicine at the University of Pennsylvania, Philadelphia, United States; [2]Department of Neuroscience, Perelman School of Medicine at the University of Pennsylvania, Philadelphia, United States; [3]Howard Hughes Medical Institute, Perelman School of Medicine at the University of Pennsylvania, Philadelphia, United States; [4]Penn Chronobiology Program, Perelman School of Medicine at the University of Pennsylvania, Philadelphia, United States

**Abstract** Sleep disturbances negatively impact numerous functions and have been linked to aggression and violence. However, a clear effect of sleep deprivation on aggressive behaviors remains unclear. We find that acute sleep deprivation profoundly suppresses aggressive behaviors in the fruit fly, while other social behaviors are unaffected. This suppression is recovered following post-deprivation sleep rebound, and occurs regardless of the approach to achieve sleep loss. Genetic and pharmacologic approaches suggest octopamine signaling transmits changes in aggression upon sleep deprivation, and reduced aggression places sleep-deprived flies at a competitive disadvantage for obtaining a reproductive partner. These findings demonstrate an interaction between two phylogenetically conserved behaviors, and suggest that previous sleep experiences strongly modulate aggression with consequences for reproductive fitness.

*For correspondence:
mattkayser@gmail.com (MSK);
amita@mail.med.upenn.edu (AS)

**Competing interests:** The authors declare that no competing interests exist.

## Introduction

Insufficient sleep impairs a wide range of essential processes such as cognition, alertness, metabolic activity, and immune function (*Foster and Wulff, 2005*). In addition, sleep disruptions influence emotional processing and can modulate affective state (*Banks and Dinges, 2007*; *Minkel et al., 2011*, *2012*). Work over many decades has suggested an interaction between sleep loss and changes in aggressive behaviors (*Kamphuis et al., 2012*), but even basic questions such as directionality of effect remain unresolved. Chronic enforced wakefulness or selective REM sleep deprivation have been linked to increased aggression in rodents, although in both cases it is not possible to distinguish the potential effect of sleep loss from prolonged physical activity and/or increased stress (*Kamphuis et al., 2012*). In contrast, work in humans indicates that while measures of irritability increase with insufficient sleep, aggression itself is unaffected or reduced (*Kamphuis et al., 2012*; *Cote et al., 2013*). While human aggression often carries a negative connotation, it can provide a competitive advantage and thereby promote survival. However, dysregulated aggression and violence are significant public health concerns (*Anderson, 2012*), as is chronic sleep insufficiency (*Czeisler, 2013*), emphasizing the need to understand how these two processes affect one another.

Both sleep and aggression are conserved across phylogeny. *Drosophila* has been established as a powerful model system for deconstructing the cellular and molecular basis of aggression, yielding novel insights into the neural basis of fighting behaviors (*Chen et al., 2002*; *Asahina et al., 2014*). Flies also exhibit sleep-like states (*Hendricks et al., 2000*; *Shaw et al., 2000*) and, in response to sleep deprivation, demonstrate deficits in behaviors like learning and memory (*Seugnet et al., 2008*). Whether sleep and aggression interact in the fly is unknown. Monoamines such as octopamine (*Crocker et al., 2010*), which is the insect analog of norepinephrine, and dopamine are potent

**eLife digest** We know from personal experience that sleepless nights can change the way we behave, sometimes making us more irritable and less adept at social interactions. However, it can be difficult to establish cause and effect: does a lack of sleep lead to altered behavior, or vice versa? The fruit fly, *Drosophila melanogaster*, is a popular model organism for studying questions like this because its neural circuitry is relatively well understood.

To explore the effects of lack of sleep on social behaviors, and in particular on aggression, Kayser et al. disrupted the sleep of male fruit flies using various techniques, such as shaking them during the night, and then observed how they behaved. The experiments revealed that sleep-deprived flies were less aggressive than flies with undisturbed sleep. Furthermore, sleep-deprived male flies were less successful at mating with female flies when they were in direct competition with a rested male fly. Normal behavior was restored when the sleep-deprived flies were allowed to recover lost sleep for as little as six hours before the next aggression assay.

To investigate how sleep loss leads to a decrease in aggressive behavior, Kayser et al. used different drugs to treat the sleep-deprived flies. A drug activating the equivalent of the noradrenergic system in flies helped them to recover normal fighting behaviors despite a lack of sleep. In mammals, noradrenaline is a chemical that affects heart rate, sleep-wake patterns, aggression and a number of other phenomena.

Although aggressive behavior is often perceived as negative in humans, it can be important for survival. Human brains and behaviors are obviously more complex than those of *Drosophila*. However, learning more about the neuronal circuits that control sleep and social behavior in fruit flies may lead to an improved understanding of these phenomena in humans and, in the longer term, the development of drugs that can influence or modulate aggressive behaviors.

regulators of arousal in *Drosophila* (*Andretic et al., 2005*; *Kume et al., 2005*; *Lebestky et al., 2009*). Distinct subpopulations of neurons of each type appear to independently modulate sleep and aggression. For octopamine, ASM neurons in the medial protocerebrum govern wake-promoting effects (*Crocker et al., 2010*), while VUM cells in the anterior or posterior brain control social behavioral choice (*Certel et al., 2010*, *2007*) or fighting behaviors (*Zhou et al., 2008*), respectively. Regarding dopamine, neurons projecting to the dorsal fan-shaped body control sleep (*Ueno et al., 2012*; *Wu et al., 2012*), and a number of distinct clusters modulate aggression, including PPM3 and T1 neurons (*Alekseyenko et al., 2013*). In sum, both of these signaling systems would be well positioned to integrate information regarding internal state and environmental demand to optimize behavioral output at a given time. Sleep serves numerous vital functions, but aggression is also a critical behavior for acquisition of food, reproduction, and predator defense (*Anderson, 2012*). Disrupting sleep processes might impair the function of neuronal controls underlying aggression.

Using *Drosophila*, we find that acute sleep deprivation strongly suppresses aggressive behaviors, while other social behaviors are unaffected. Reduced aggression occurs with different forms of sleep deprivation and is reversible with sufficient recovery sleep. Pharmacologic experiments reveal that an octopamine agonist specifically restores aggression in sleep-deprived flies, and we use genetic approaches to suggest that sleep loss itself, rather than manipulation of aggression neurons, is required for changes to aggression. Finally, we demonstrate that sleep deprived flies are at a disadvantage for reproductive success when competing against flies whose sleep has been unperturbed.

## Results

### Acute sleep deprivation suppresses aggression

To test how sleep deprivation affects aggressive behaviors in *Drosophila*, we focused on 4–7 day old Canton-S (CS) males in social isolation since shortly after eclosion. Flies were acutely sleep deprived for 12 hr overnight using mechanical stimulation, and aggression assays were performed the following morning, either by pairing flies within the same condition or pairing one control fly with a sleep-deprived fly ('between conditions', for which males of each condition were tracked with a small dot of

paint on the thorax). We found that acute sleep deprivation resulted in a profound suppression of aggression in both cases (*Figure 1A*; *Figure 1—source data 1*). In assays between deprived and non-deprived flies, non-deprived flies showed frequent lunges and were nearly always dominant over deprived flies (*Figure 1B*). In contrast, flies that were sleep deprived overnight rarely lunged, even when attacked by a non-deprived fly, suggesting suppression of reactive aggression by sleep deprivation. In assays between 2 sleep-deprived flies, the males engaged in aggressive behaviors significantly less often than pairs of control flies (*Figure 1A*; *Videos 1, 2*). The reduced aggression within condition did not simply derive from lack of social encounters, as sleep-deprived and control pairs spent similar amounts of time interacting during the beginning of the assay (*Figure 1—figure supplement 1*); sleep-deprived flies could also be observed throughout the assay in close proximity without engaging in fighting behaviors, which rarely occurred in control flies. When lunging did occur, the latency to first lunge in pairs of sleep-deprived flies following initial social encounter was markedly increased compared to control flies (*Figure 1C*), and dominance was rarely established. Together, these results suggest deficits in both proactive and reactive aggression, and demonstrate that sleep deprivation negatively impacts fighting behaviors.

We next tested whether the link between sleep and aggression is specific to sleep loss during the night. The CS males used in our experiments show high sleep amounts during both day (450.77 ± 77.09 min) and night (528.91 ± 109.74 min; *Figure 1—figure supplement 2*), allowing us to test day and night sleep requirements. Mechanical sleep deprivation for 11 hr during the day with fighting assays performed during the final hour of the light period also reduced aggression (*Figure 1A*) and shifted dominance towards non-deprived flies (*Figure 1B*), indicating that timing of sleep loss to day or night is not critical. How much sleep loss is required to affect subsequent fighting behaviors? We mechanically sleep deprived flies for the final 1, 3, or 6 hr(s) of the night, followed by aggression assays the next morning. Focusing on assays within condition, we found that 6 hr of sleep loss resulted in suppression of fighting (*Figure 1D*) similar to 12 hr of sleep loss. However, sleep deprivation for the final 1 or 3 hr(s) of the night caused no reduction in fighting behavior (*Figure 1D*), consistent with the timeframe of sleep loss required for learning deficits in flies (*Seugnet et al., 2008*). To control for the possibility that mechanical stimulation/fatigue and not sleep loss per se is responsible for reduced aggression, we mechanically stimulated flies for a 6 hr period spanning the final 3 hr of night and first 3 hr of day. During this time, flies exhibit more wake in comparison to the final 6 hr of the night, only losing ~3 hr of sleep (sleep lost = 186.72 ± 61.03 in night/day transition; sleep during final 6 hr of night = 293.41 ± 61.49). Continuous stimulation for 6 hr during the night/day transition period did not affect subsequent fighting (*Figure 1E*), indicating that sleep deprivation and not mechanical stimulation is responsible for suppression of aggression.

To rule out the possibility that acute sleep deprivation caused permanent impairment of aggressive behaviors, we examined lunging after 12 hr of overnight mechanical sleep deprivation followed by a 24 hr recovery period. Fighting returned to baseline levels with 1 day recovery after sleep loss (*Figure 1A*), and dominance was equally likely for a previously sleep-deprived or non-deprived fly when paired against one another (*Figure 1B*). Thus, neither sleep deprivation, nor the mechanical stimulation used to deprive, result in injury to the fly or long-lasting impairment. Next, we further investigated the timeframe for recovery following 12 hr of mechanical sleep deprivation. Deprived or control flies were assessed within condition at 1, 3, 6, or 9 hr after overnight sleep deprivation (*Figure 1F*). Following 1 and 3 hr of recovery sleep, aggression was still markedly suppressed. By 6 hr we observed recovery of fighting behaviors, with lunge counts no longer significantly reduced compared to the 6 hr control group. 9 hr after the end of sleep deprivation, levels of aggression remained indistinguishable from controls. Consistent with previous work (*Hoyer et al., 2008*), we detected no differences in baseline aggression in control groups throughout the day (*Figure 1F*).

Do other methods of sleep deprivation similarly influence aggression? We noted sleep in CS males to be exquisitely sensitive to high temperatures, with near total sleep loss when exposed to 31°C overnight. Temperature-dependent sleep deprivation also resulted in reduced lunging with aggression assays performed the following morning at 25°C (*Figure 1—figure supplement 3*), indicating that suppressed aggression is not specific to mechanical sleep deprivation. To examine whether sleep deprivation in the fly indiscriminately impairs other complex motor programs, we tested if courtship between a socially naive male and virgin female is affected. Consistent with previous work (*Seugnet et al., 2011*; *Kayser et al., 2014*), 12 hr of mechanical sleep deprivation during the night had no effect on courtship behavior the following morning as measured by courtship

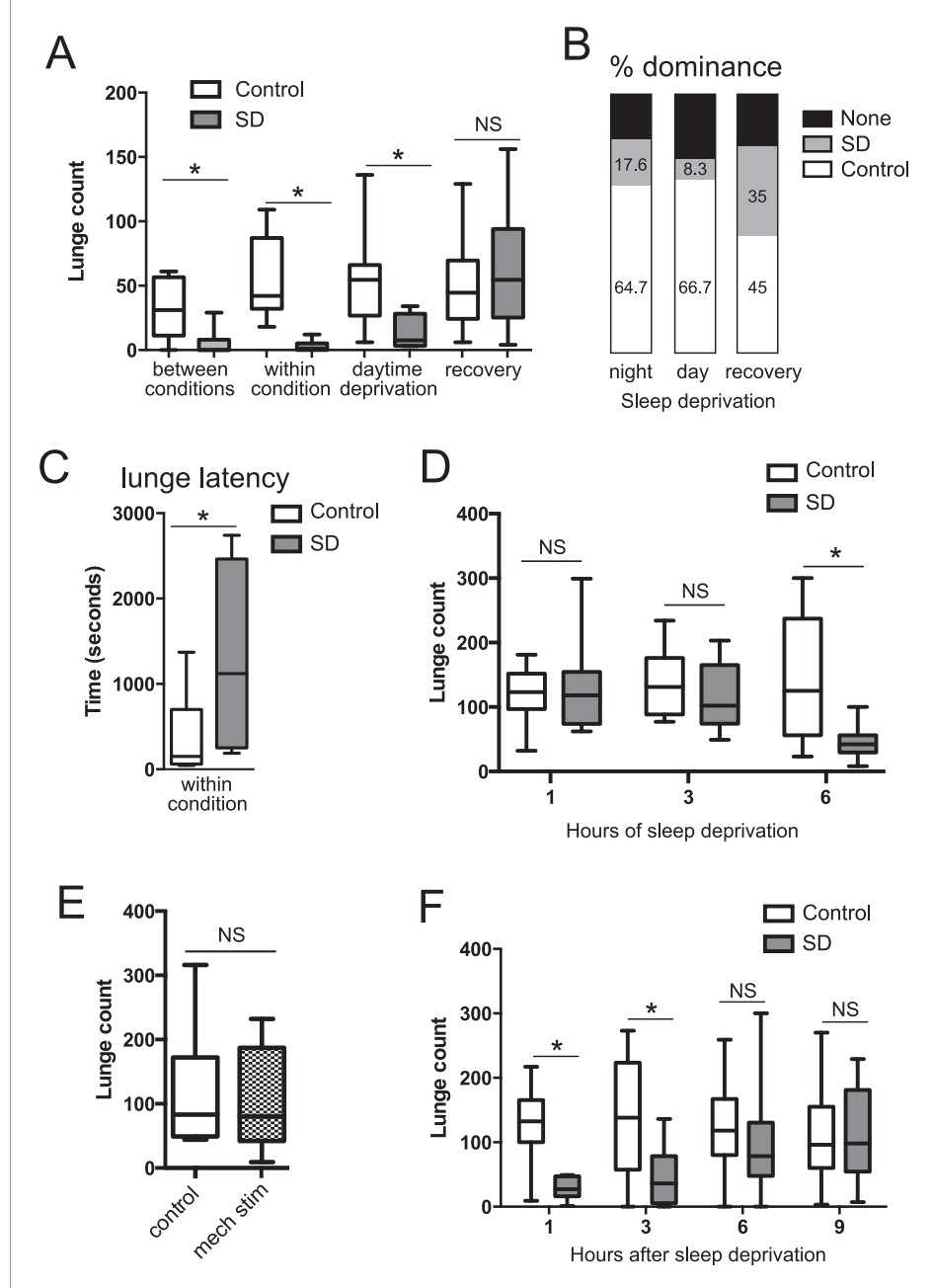

**Figure 1**. Acute sleep deprivation suppresses aggressive behaviors. (**A**) Quantification of aggression (lunge count) in control or sleep deprived male CS flies. 'Between conditions' indicates fights between a control fly and a sleep-deprived fly; 'within condition' is a fight between 2 sleep-deprived or 2 control flies. 'Daytime deprivation' and 'recovery' refers to fights between conditions (n = 16, 16, 9, 10, 10, 8, 14, 9 from left to right). (**B**) Percentage of flies in each condition establishing dominance (fights between conditions). (**C**) Latency to first lunge following first social encounter (fights are within condition; n = 9 control, 7 deprived). (**D**) Lunging follow sleep deprivation during the final 1, 3 or 6 hr(s) of the night (fights within condition; n = 8, 8, 13, 9, 12, 12 from left to right). (**E**) Lunging following 6 hr of mechanical stimulation during the final 3 hr of night and first 3 hr of day (fights within condition; n = 11 for both). (**F**) Recovery of aggression following prior sleep deprivation for 12 hr (fights within condition; n = 10, 10, 18, 17, 19, 18, 15, 12 from left to right). Box plots in this figure and all others represent median value (horizontal line inside box), interquartile range (height of the box, 50% of the data within this range), and minimum and maximum value (whiskers). Bar graphs in this figure and all others are presented as mean ± s.e.m. *p < 0.05; 1 way ANOVA with Tukey's (**A**, **D**) or Sidak's (**F**) post-hoc test; unpaired two-tailed Student's t-test (**C**, **E**). SD = sleep deprivation.
*Figure 1. continued on next page*

*Figure 1. Continued*

The following source data and figure supplements are available for figure 1:

**Source data 1**. Quantification of lunging following mechanical sleep deprivation.

**Figure supplement 1**. Quantification of percentage of time pairs of flies spent interacting, with or without prior sleep deprivation (n = 12 pairs for control and deprived).

**Figure supplement 2**. Sleep trace of CS male, showing high sleep amounts during both day and night (n = 22 flies).

**Figure supplement 3**. Quantification of aggression following 12 hr of high temperature (31°C) -induced sleep deprivation (fights within condition at 25°C; n = 21 control, 28 deprived).

**Figure supplement 4**. Courtship index and copulation frequency of control or mechanically sleep-deprived males (n = 15 control and deprived).

index or copulation frequency over a 10 min period (*Figure 1—figure supplement 4*). Thus, a distinct complex behavior remains intact following acute sleep loss, suggesting that like learning and memory, aggression is a specific behavior that is impaired following sleep deprivation.

## Bifunctional role of octopamine in sleep and aggression

Octopamine and dopamine have been implicated in controlling both sleep/arousal and aggressive behaviors. We examined whether either of these monoamines play a role in coupling sleep deprivation to changes in aggression. First, we asked how thermogenetic sleep deprivation via overnight activation of octopaminergic or dopaminergic neurons affects next day fighting. The *Drosophila* thermosensitive cation channel *dTrpA1* (*Hamada et al., 2008*) was expressed using *Tdc2-GAL4* (octopamine neurons) or *TH-GAL4* (dopamine neurons), and flies were exposed to 29°C for 12 hr overnight. Activation of neurons using either GAL4 line resulted in near total sleep deprivation (*Figure 2—figure supplement 1*); importantly, overnight exposure to 29°C in flies lacking either the GAL4 or UAS did not impact sleep (*Figure 2—figure supplement 2*), in contrast to CS males exposed to 31°C. Aggression was assessed the following morning at 23°C, below the threshold for TrpA1 activation. Sleep deprivation by activation of Tdc2+ or TH+ neurons caused a significant suppression of aggression the following day (*Figure 2A*; *Figure 2—source data 1*), while genetic and temperature controls with normal prior sleep showed no such effect (*Figure 2A*; *Figure 2—figure supplement 2*). Our results suggest that perturbation of octopamine and/or dopamine signaling as a result of sleep deprivation could impair aggressive behaviors. Using a pharmacologic approach we tested whether octopamine or dopamine agonists rescue reduced aggression. Flies were fed the octopamine agonist chlordimeform (CDM) or the dopamine agonist L-DOPA leading up to aggression assays that were preceded by thermogenetic sleep deprivation. CDM, but not L-DOPA, rescued aggression following sleep deprivation via activation of Tdc2+ neurons (*Figure 2B*). Surprisingly, CDM also rescued aggression following sleep deprivation via activation of TH+ neurons, while L-DOPA did not (*Figure 2C*). This effect is not specific to thermogenetic sleep deprivation, as CDM alone rescued aggression after mechanical sleep deprivation as

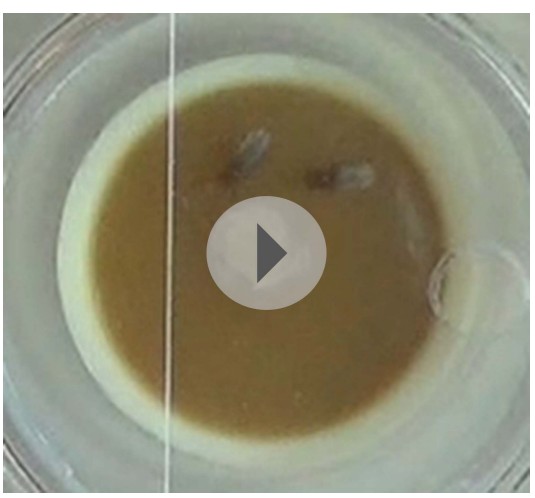

**Video 1.** Aggression assay between two control flies (on regular food with a drop of yeast paste in the center).

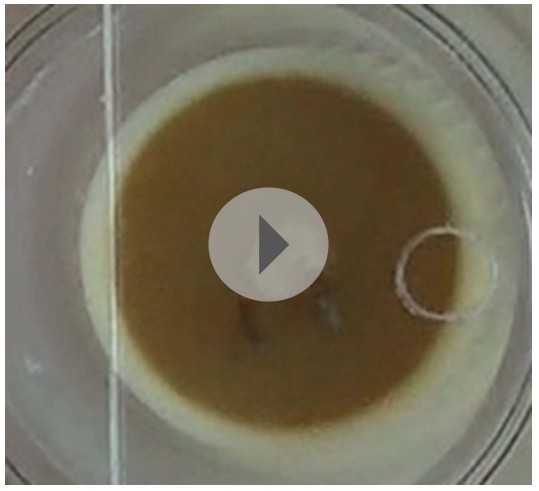

**Video 2.** Aggression assay between two sleep-deprived flies.

well (**Figure 2—figure supplement 3**). These results suggest that aggression-relevant octopamine function is compromised downstream of sleep deprivation signals, regardless of method of sleep loss.

To rule out the possibility that CDM is simply a more potent inducer of fighting behaviors, we investigated the effect of both drugs on baseline aggression and found no increase in fighting following either 1 or 3 days of drug feeding (**Figure 2—figure supplement 4**). Previous work has shown that social experience through group housing of flies also dramatically reduces aggression (**Wang et al., 2008**; **Zhou et al., 2008**), and we examined if a shared mechanism suppresses fighting in both cases. In contrast to sleep deprivation, reduced aggression secondary to group housing was similarly rescued either by CDM or L-DOPA (**Figure 2D**), indicating that the drugs can function comparably to rescue aggression depending on the method of suppression. Moreover, while sleep deprivation and social experience both suppress aggression, they appear to do so through distinct though potentially overlapping mechanisms.

Does sleep deprivation itself suppress aggression, or does reduced fighting stem from decreased locomotor activity after sleep loss? Following thermogenetic sleep deprivation we quantified motor activity. As expected, sleep-deprived flies were less active than non-deprived controls (**Figure 2E**), presumably due to increased homeostatic sleep drive. While only CDM rescued aggressive behaviors following sleep deprivation (**Figure 2B**), we found that both CDM and L-DOPA rescued locomotor activity back to control levels (**Figure 2E**). Together, these experiments dissociate motor activity from aggression following sleep deprivation, and indicate that sleep loss itself impairs aggressive behaviors.

A group of 2–4 Tdc2+ neurons in the posterior fly brain within the VUM cluster near the suboesophageal (SOG) ganglion have been implicated in the role of octopamine modulation of aggression (**Zhou et al., 2008**). These neurons can be targeted genetically using *Tdc2-GAL4* with the *Cha-GAL80* suppressor (**Figure 3A**; **Figure 3—source data 1**) (**Zhou et al., 2008**). Suppressed aggression following overnight activation of all Tdc2+ neurons with TrpA1 might be independent of sleep deprivation, and derive from hyperstimulation and subsequent quiescence of the Tdc2+ Cha- aggression cells. To rule out this possibility, we activated these neurons for 12 hr overnight at 29°C, followed by next day aggression assays at 23°C. Activation of Tdc2+ Cha- neurons did not cause sleep deprivation (**Figure 3B**), consistent with research localizing the wake-promoting octopamine neurons to dorsal brain regions (**Crocker et al., 2010**). Importantly, sustained activation of Tdc2+ Cha- neurons also did not impair aggression the following day (**Figure 3C**). Together these results show that reduced aggression following overnight activation of Tdc2+ neurons is not caused by hyperstimulation of Tdc2+ Cha- neurons, and further suggest that sleep deprivation per se is required for suppression of aggression.

## Impaired reproductive fitness following sleep deprivation

In addition to a role in promoting aggression, octopamine also modulates the choice between courtship and aggression in males (**Certel et al., 2010**, **2007**). A distinct subpopulation of VUM octopamine neurons that co-express the male variant of the sex determination factor Fruitless (Fru(M)) in the anterior SOG have been implicated in this behavioral choice (**Certel et al., 2010**, **2007**). We assessed whether sleep deprivation might influence social behavior decision-making in addition to aggression itself. Either 2 sleep-deprived or 2 control male CS flies (4–7 days old; reared in isolation) were placed with a virgin CS female, and we determined percent time spent courting the female in comparison to the other male. Previous work has shown that disturbing the function of Tdc2+ Fru(M)+ neurons results in abnormal elevation of male–male courtship, whereas near exclusive male-female courtship with intermale aggression is found under normal conditions (**Certel et al., 2010**, **2007**).

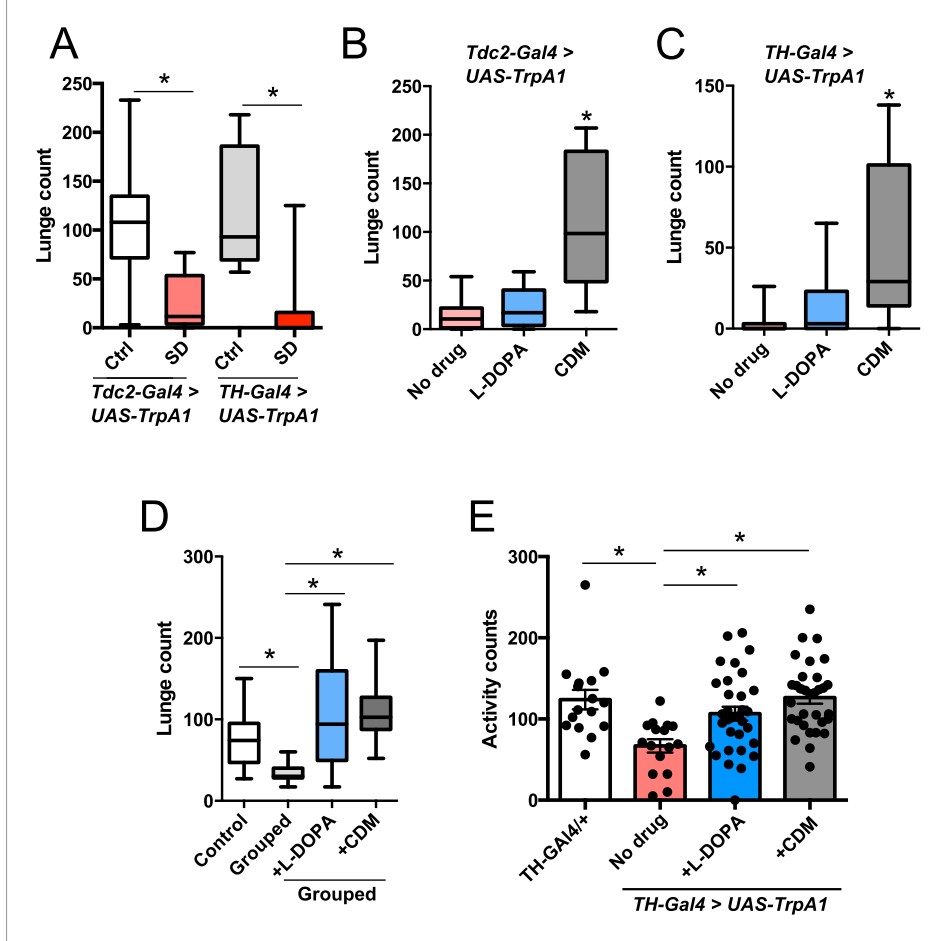

**Figure 2**. Octopamine agonist CDM rescues reduced aggression following sleep deprivation. Quantification of aggression at 23°C in *Tdc2-GAL4>UAS-TrpA1* or *TH-GAL4>UAS-TrpA1* males that were thermogenetically sleep-deprived the prior night at 29°C (SD, red bars) or remained at 21°C (Ctrl, white/gray bars) (n = 10, 16, 10, 12 from left to right). (**B**, **C**) Rescue of suppressed aggression in *Tdc2-GAL4>UAS-TrpA1* (**B**) or *TH-GAL4>UAS-TrpA1* (**C**) males fed either CDM or L-DOPA and thermogenetically sleep-deprived (**B**, n = 12, 10, 10; **C**, n = 12, 12, 12 from left to right). (**D**) Rescue of suppressed aggression in group-housed CS males fed either CDM or L-DOPA compared to males reared in isolation (Control) (n = 9, 9, 12, 12 from left to right). (**E**) Locomotor activity over 1 hr following exposure to 29°C the prior night (n = 16, 16, 32, 32 from left to right). All fights are within condition. *p < 0.05; 1 way ANOVA with Sidak's (**A**) or Tukey's (**B**–**E**) post-hoc test.

The following source data and figure supplements are available for figure 2:

**Source data 1**. Quantification of lunging following thermogenetic sleep deprivation and pharmacologic rescue.

**Figure supplement 1**. Sleep traces in *TH-GAL4>UAS-TrpA1* (n = 24 red, 20 black) or *Tdc2-GAL4>UAS-TrpA1* (n = 10 red, 10 black) flies loaded at ZT6 with temperature shift (red trace) to 29°C at ZT12, compared to controls (black trace) remaining at 21°C.

**Figure supplement 2**. (Left) Sleep traces in *Tdc2-GAL4>UAS-TrpA1* (black) and GAL4 (red) and UAS (blue) controls loaded at ZT6 with temperature shift (pink box) to 29°C at ZT12.

**Figure supplement 3**. Quantification of aggression in CS males fed either CDM or L-DOPA and mechanically sleep-deprived (n = 17, 20, 24 from left to right).

**Figure supplement 4**. Quantification of aggression following 1 days or 3 days of drug exposure in isolated males.

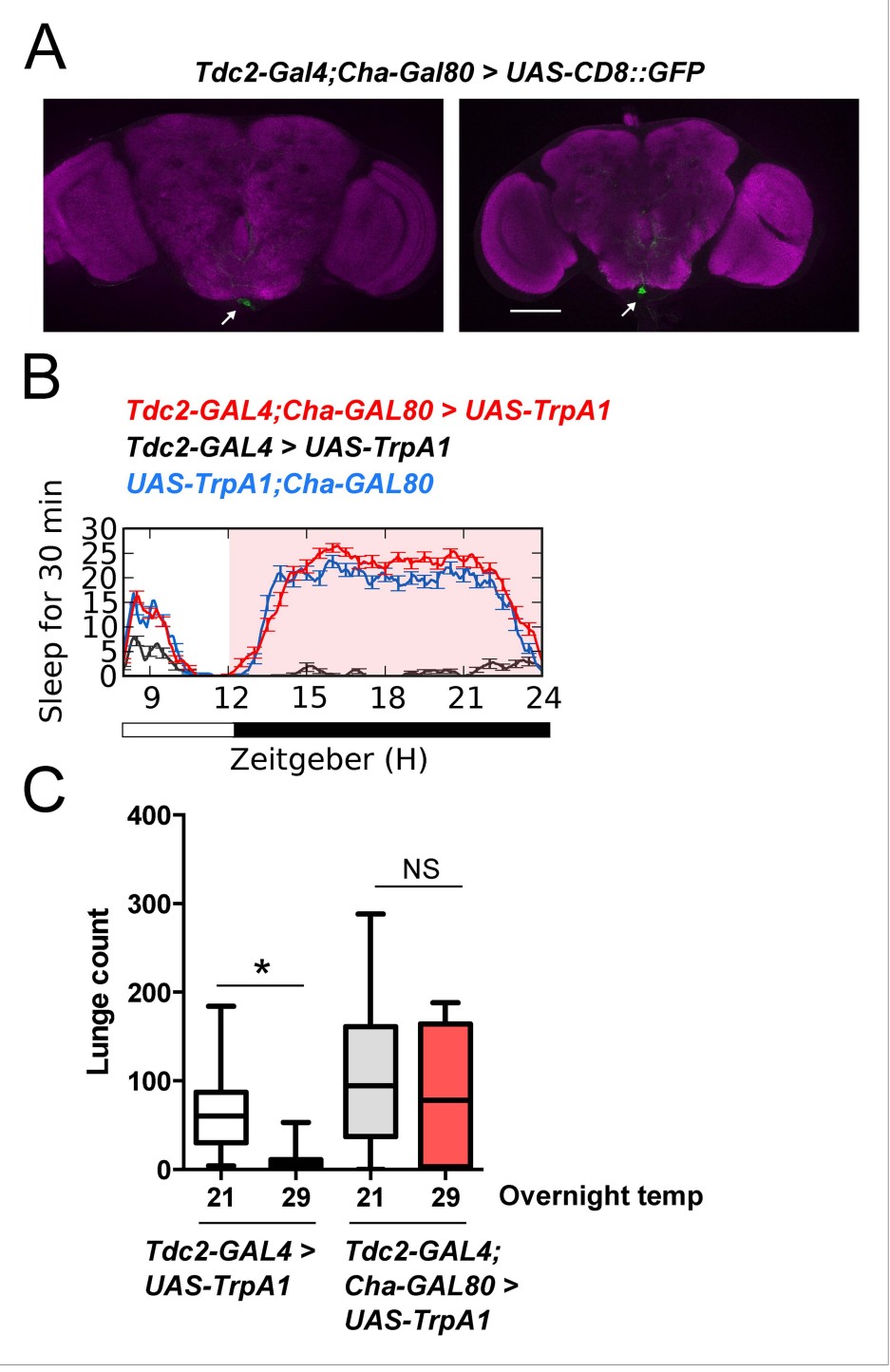

**Figure 3**. Sleep loss is required for suppressed aggression following octopaminergic activation. (**A**) Images of Tdc2+ Cha- neurons in brains from *Tdc2-GAL4;Cha-GAL80>UAS-CD8::GFP* flies immunostained for GFP (green) and nc82 (magenta). Arrows indicate 2–4 VUM neuron cluster in posterior brain. Scale bar = 100 μm. (**B**) Sleep traces in *Tdc2-GAL4;Cha-GAL80>UAS-TrpA1* (red), *Tdc2-GAL4>UAS-TrpA1* (black), and *UAS-TrpA1;Cha-GAL80* (blue) flies with temperature shift (pink box) to 29°C at ZT12 (n = 12 flies for all conditions). (**C**) Quantification of aggression in *Tdc2-GAL4>UAS-TrpA1* males exposed to elevated temperature (and sleep deprived) overnight compared to *Tdc2-GAL4;Cha-GAL80>UAS-TrpA1* males, and temperature controls (fights within condition; n = 14, 15, 16, 15 flies, from left to right). *p < 0.05; 1 way ANOVA with Sidak's post-hoc test.

*Figure 3. continued on next page*

*Figure 3. Continued*

The following source data is available for figure 3:

**Source data 1**. Quantification of lunging following thermogenetic activation of Tdc2+Cha- neurons.

Following mechanical sleep deprivation overnight, we found that flies overwhelmingly courted females, with no difference in the minimal time engaged in male–male courtship compared to pairs of non-deprived control males (*Figure 4A,B*; *Figure 4—source data 1*). Thus social behavioral choice is normal between pairs of sleep-deprived flies.

In the behavioral choice assays, we also noticed what appeared to be decreased competition for the female target between sleep-deprived flies in comparison to non-deprived control conditions. Non-deprived flies were frequently observed to interrupt courtship activities of the other male, and at times, engage in fighting behaviors away from the female altogether. These behaviors were less frequently observed between sleep-deprived males. To quantify the reduced male–male competition for a female mate in sleep-deprived flies, we determined the percent of time that the 2 males and 1 female were observed clustered together or in a 'chain' formation, with both males actively courting the female concurrently. During these periods, males were clearly observed to engage in low intensity fighting behaviors like fencing, which sometimes escalated to brief high intensity tussling or lunging. We found that in non-deprived flies, 26% of time in the assay was spent in chains or clusters; this was reduced to 17.2% in pairs of sleep-deprived males (*Figure 4C*; *Videos 3, 4*), suggesting that competitive social interactions are reduced following sleep deprivation. Taken together, our results demonstrate that while social behavioral choice is unaffected, impairments in aggression influence how males competitively court a female.

Success in aggressive interactions has been correlated with fitness for mating (*Dow and von Schilcher, 1975*) (though not in flies bred to be hyper-aggressive [*Dierick and Greenspan, 2006*; *Penn et al., 2010*]). Does reduced aggression following sleep deprivation have a functional consequence on reproductive fitness in a competitive environment? We paired control and mechanically sleep-deprived CS males together with a virgin female and assayed copulation success. Males of each condition were tracked with a dot of paint on the thorax. In these assays, we found that control flies were more likely to 'win' copulation with the female compared to sleep deprived flies (~63% vs 37%; *Figure 4D*). Thermogenetic sleep deprivation via activation of Tdc2+ neurons overnight similarly impaired next-day performance in the competitive copulation assay against a genetically identical male (*Figure 4E*). We then tested whether competitive copulation deficits can be reversed with rescue of aggression after sleep deprivation. *Tdc2-GAL4>UAS-TrpA1* males were fed CDM or L-DOPA and sleep deprived overnight by exposure to 29°C. Competitive copulation assays were performed the following morning at 23°C against non-deprived *Tdc2-GAL4>UAS-TrpA1* males that received no drug. Consistent with rescue of aggressive behaviors, we found that CDM restored copulation success of sleep-deprived males competing against controls, while L-DOPA had no effect (*Figure 4E*). Neither drug had a significant effect in assays against controls (no drug) in the absence of sleep manipulations (*Figure 4—figure supplement 1*). These results suggest that suppression of aggression following sleep loss is deleterious when competing with a non-deprived male for a reproductive partner.

## Discussion

Here we show that sleep deprivation reduces aggression in *Drosophila* while other complex social behaviors are unimpaired. The suppression of aggression is independent of the method used to induce sleep loss and reversible with sufficient recovery sleep. We provide evidence that octopamine plays a role in mediating reduced aggression levels following sleep loss, and we find using competitive courtship assays that sleep deprivation places flies at a disadvantage for reproductive success due to insufficient aggression towards competing males. These results demonstrate that aggression is a specific social behavior that becomes dysregulated by sleep deprivation.

Subjective experience in humans indicates that irritability increases with insufficient sleep, which is supported by validated measures of emotional reactivity and mood (*Banks and Dinges, 2007*); however, irritability and aggression are likely dissociable in higher organisms. While there are well-validated measures of aggression as a lifetime personality trait, tools to measure changes to

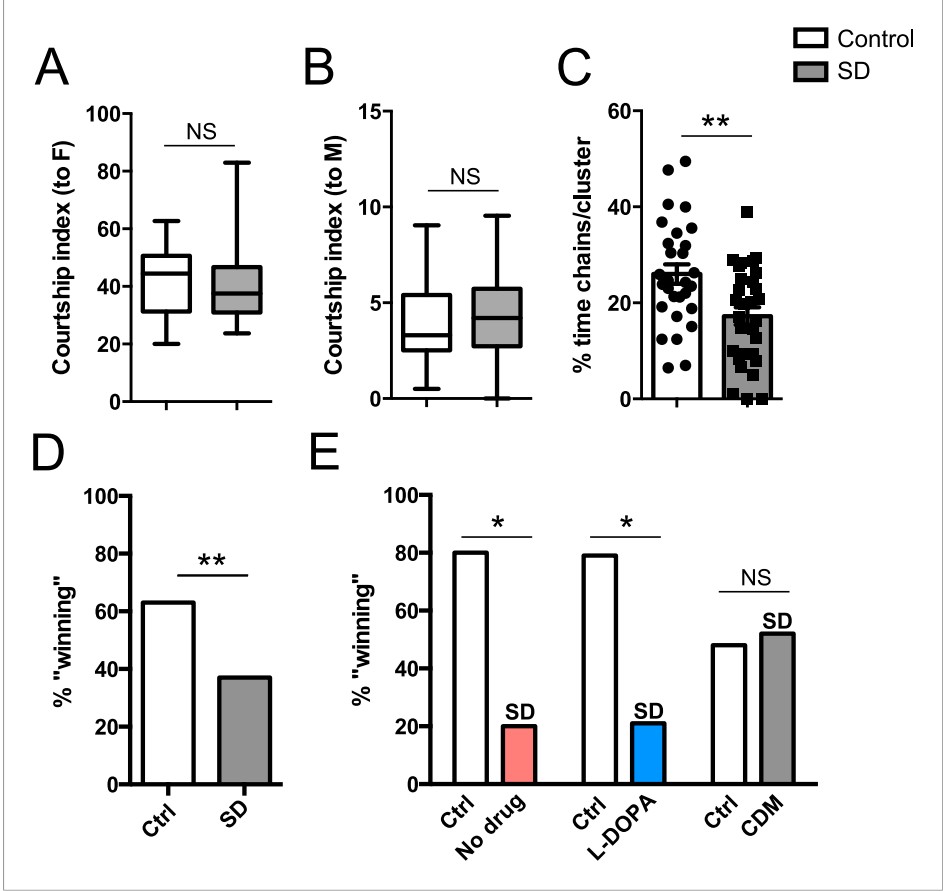

**Figure 4**. Suppressed aggression following sleep deprivation impairs reproductive fitness. Courtship index towards the female target (**A**) or male (**B**) in control (white, n = 32) or sleep-deprived (gray, n = 31) flies during a competitive courtship assay (within condition). (**C**) Percentage of time spent in chains/clusters during the competitive courtship assay with control (n = 29) or sleep-deprived (n = 30) flies (within condition). (**D**) Percentage of assays in which the control or sleep-deprived male first copulates with ('wins') the female target in a competitive courtship assay between conditions (n = 116 assays, 5 independent experiments). (**E**) As in (**D**) but with control vs sleep-deprived ± drug rescue males (control vs SD + no drug, n = 20 assays; control vs SD + L-DOPA, n = 28 assays; control vs SD + CDM n = 27 assays; 3 independent experiments). **p < 0.01, *p < 0.05; Unpaired two-tailed Student's t-test (**A–C**) or two-tailed Binomial test (**D**, **E**).

The following source data and figure supplement are available for figure 4:

**Source data 1**. Courtship and competitive copulations measures following sleep deprivation.

**Figure supplement 1**. Percentage of assays in which the control or drug condition male first copulates with the female target in a competitive courtship assay between conditions (control vs L-DOPA, n = 30 assays; control vs CDM n = 30 assays; 3 independent experiments).

aggression in humans over relatively short-time periods (hours to days) are less well-established, as is measuring irritability in animal models independent of overt behavioral expression. Most human studies of the interaction between sleep and aggression have been correlational and utilized self-reported measures (*Kamphuis et al., 2012*). Nonetheless, more recent human work is informative regarding dissociation of irritability and/or impulsivity from aggression itself: research supports a reduced capacity to inhibit impulsivity to negative stimuli after sleep deprivation (*Anderson and Platten, 2011*), while a study designed to measure reactive aggression in sleep-deprived men reported diminished aggression in response to provocation (*Cote et al., 2013*). The rodent literature supports the contrasting hypothesis that sleep insufficiency leads to *increased* aggression

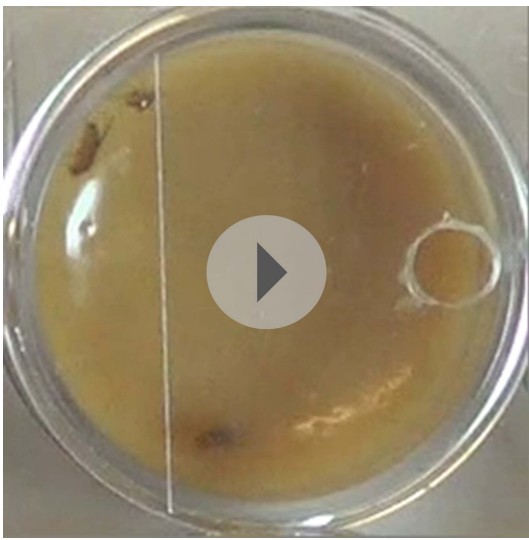

**Video 3.** Competitive courtship assay between 2 control males (white or red painted dot) and a virgin female.

(*Kamphuis et al., 2012*). These differences might result from confounds such as the impact of sustained activity or stress rather than sleep deprivation per se, or REM sleep loss vs total sleep loss. Regardless, while the repertoire of emotional experience and output in humans is far more complex than in a laboratory animal, the central effect of sleep deprivation on aggression appears to be conserved between flies and humans.

Segregating stress effects from sleep loss itself when sleep-depriving animals—including humans—presents obstacles. Using multiple types of sleep deprivation stimuli combined with genetic manipulations, our data suggest that the impact of sleep deprivation on aggression is causally related to sleep loss. Nonetheless, experimental paradigms that prevent sleep in the face of heightened sleep need are inherently stressful to an animal. The specificity with which aggressive social interactions are impaired as opposed to all social behaviors serves as a strong indicator that the effects we observe are not simply part of a broader stress response; more-over, other stressors are known to negatively impact courtship behaviors (*Patton and Krebs, 2001*; *Christie et al., 2013*), suggesting that courtship is not uniquely immune to stress-related impairment.

Aggression in *Drosophila* has long been appreciated to play a role in mate selection (*Sturtevant, 1915*; *Dow and von Schilcher, 1975*). Male flies fight more in the presence of a female, and recent work has shown that prior exposure to a female suppresses this effect (*Yuan et al., 2014*); the aforementioned study focused on aggressive behaviors occurring after male flies copulate with females in the arena, suggesting that aggression may function both to obtain a reproductive partner and possibly guard that partner after mating (*Yuan et al., 2014*). Our data indicate that even in the setting of sleep deprivation, male flies will successfully court a female partner in the absence of competition. However, when competing for a female mate against a non-deprived control male, sleep deprivation impairs sexual fitness, likely due to reduced aggression towards the other male: pharmacologic restoration of aggression with an octopamine agonist likewise rescues sexual fitness. Importantly, reduced performance in the competitive copulation assay occurs in males that were isolated since eclosion, eliminating the possibility that previously-established hierarchical cues are involved and emphasizing the crucial role of sleep for normal function.

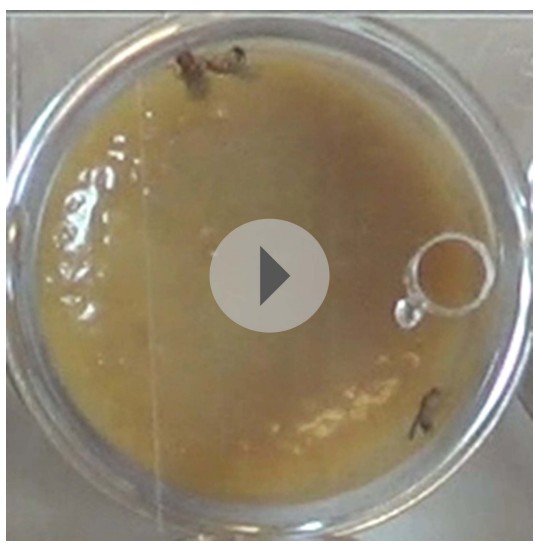

**Video 4.** Competitive courtship assay between 2 sleep-deprived males (white or red painted dot) and a virgin female.

Sleep loss is required for subsequent suppressed aggression, while prolonged prior excitation of aggression-relevant neurons has no effect on fighting the following day. How is sleep need conveyed to aggression loci? Our results implicate octopaminergic signaling in this process, and suggest that neurons involved in setting aggression levels, but not male choice between courtship and aggressive behaviors, are specifically impacted: while intermale aggression

is reduced with sleep deprivation, intermale courtship is not, by default, increased. Norepinephrine, the mammalian analog of insect octopamine, has a critical role in aggression in vertebrates; noradrenergic neurons are also important regulators of sleep-wake transitions. Identification of specific cellular subpopulations whose output is altered by sleep deprivation in *Drosophila* will enable mapping the neural circuits that relay sleep information to aggression centers, as well as investigation of the molecular signals that control deprivation-dependent changes to octopamine aggression neurons. The conserved cellular substrates of these behaviors between flies and mammals suggest such findings will be of relevance to understanding the neurobiological basis of aggression and impairment after sleep loss. In sum, our results indicate that two innate behaviors—sleep and aggression—are coupled, and suggest that molecular signals generated by sleep deprivation might be potential targets for selective modification of aggressive behaviors.

## Materials and methods

### Fly stocks

*TH-GAL4*, *Tdc2-GAL4*, *Cha-GAL80* are from laboratory stocks, and were outcrossed 8× into a *w1118; CS* background (gift from D Anderson). Wild-type CS flies are a gift from E. Kravitz. *UAS-dTrpA1 (in w1118;CS background)* are a gift from D Anderson. Flies were maintained in bottles on standard food at 25° on a 12 hr:12 hr LD cycle.

### Sleep/locomotor assays

For sleep experiments, flies were loaded into glass tubes containing 5% sucrose and 2% agar at ~ ZT6-8 by gentle aspiration (if being used for a behavioral assay the following day). Locomotor activity was monitored using the *Drosophila* Activity Monitoring (DAM) system (Trikinetics, Waltham MA). Activity was measured in 1 min bins and sleep was identified as 5 min of consolidated inactivity (*Hendricks et al., 2000*; *Shaw et al., 2000*). Data was processed using PySolo (*Gilestro and Cirelli, 2009*). Mechanical sleep deprivation was accomplished using a Trikinetics vortexer mounting plate, with shaking of monitors for 2 s randomly within every 20 s window for 12 hr during the night. Temperature-dependent sleep deprivation in CS males was at 31°C during the night; thermogenetic sleep deprivation was at 29°C. For *Figure 2E*, locomotor activity was quantified as total number of beam breaks from ZT1-2.

### Aggression assays

Flies were moved into isolation tubes (5 ml tubes, Falcon 352002) shortly after eclosion unless otherwise specified. Following sleep assays in DAM tubes, flies recovered for 30 min in isolation on regular food in isolation tubes, and then were moved to aggression arenas by gentle aspiration. Aggression assays were performed and scored as previously described (*Chen et al., 2002*; *Certel and Kravitz, 2012*) with minor modifications. Assays consisted of fights between 2 males in 1 well of a 12 well plate with a food cup in the center. Yeast paste or a buried headless female were placed in the center of the cup, which was well lit. The sides of each arena were coated in Fluon (Bioquip Products, Rancho Dominguez, CA) and lid with Rain-X (ITW Global Brands, Houston, TX). Fights were recorded from above using a video camera (Sony HDR-CX210) and lunges scored manually, blind to condition. For fights between conditions (and all competitive copulation experiments) male flies were labeled with a small dot of acrylic paint at least 24 hr prior to the assay; the paint color for each condition was randomized between experiments. Lunge count for 'between condition' assays (control vs SD) was quantified separately for each fly of the designated condition; lunge count 'within condition' was quantified as the combined number of lunges from both flies in the assay. Dominance (scored only in 'between condition' assays) was determined as repeated lunges by one fly followed by retreat of the other to the edge of the cup or off the cup altogether (*Alekseyenko et al., 2014*). All assays were performed for 30 min at 25°C and 40–60% humidity unless otherwise specified. For *Figure 1A*, social interaction was defined as time spent with the flies within 1 body length of one another. Interaction was quantified during the first 5 min of the assay or until first lunge, whichever occurred first. For pharmacologic rescue experiments, flies were fed chlordimeform (CDM; 0.05 mg/ml; Sigma) or L-DOPA (3 mg/ml; R&D Systems) mixed into 5% sucrose and 2% agar beginning ~4 hr prior to lights off, and continued to feed on the drug throughout the night (during 12 hr of thermogenetic sleep

deprivation or mechanical sleep deprivation during the final 6 hr of night). At ZT0 (end of sleep deprivation), flies recovered for 1 hr on drug/control mixed into regular food at the same concentration prior to aggression assays at ZT1. For group-housing experiments, males were raised in groups of 10 flies for 4–5 days, then isolated on drug/control for 20 hr prior to assay at ZT1.

## Courtship assays

Virgin male flies were collected shortly after eclosion and housed in isolation. Female CS virgins (3–7 days post eclosion) were used in all courtship assays. Following sleep assays, flies recovered for 30 min in isolation on regular food, and then a male and female were gently aspirated into a well-lit porcelain mating chamber (25 mm diameter and 10 mm depth) covered with a glass slide. Courtship index (CI) was determined as the percentage of total amount of time a male was engaged in courtship activity during a period of 10 min or until successful copulation. Copulation frequency was calculated as percentage of flies in each condition that successfully copulated during the 10 min assay. For 'within condition' competitive courtship assays (*Figure 4A–C*), 2 males and 1 virgin female were loaded into 1 well of a 12 well plate containing 5 ml of food (*Certel et al., 2010*, *2007*). Courtship and copulation measures were determined for the male first demonstrating sustained courtship (>20 s) towards the female. Assays were recorded and scored blind to experimental condition. For competitive courtship assays between conditions, one male fly of each condition (control or deprived) and 1 virgin female were aspirated into a well, and the male that first copulated with the female was determined the 'winner'. Experiments were scored blind to condition.

## Immunohistochemistry and imaging

Brains were dissected in PBS, fixed in 4%PFA for 30 min at room temperature, and cleaned of remaining tissue in 0.3% PBST. Following 3 × 10 min washes in PBST, brains were blocked in 5% normal donkey serum (NDS) and incubated with primary antibody in block at 4° overnight. Following 3 × 10 min washes in PBST, brains were incubated with secondary antibody in block for 2 hr at room temperature. Following 3 × 10 min washes in PBST, brains were mounted in vectashield. Primary antibodies included: Mouse anti-nc82 (1:1000, Developmental Studies Hybridoma Bank), Rabbit anti-GFP (1:1000, Molecular Probes). Secondary antibodies included: FITC donkey anti-rabbit (1:500, Jackson), Cy5 donkey anti-mouse (1:500, Jackson). Brains were visualized with a TCS SP5 confocal microscope and images processed in NIH ImageJ.

## Statistical analysis

All analysis was done in GraphPad (Prism). Individual tests and significance are detailed in figure legends.

## Acknowledgements

We thank D Raizen, P Meerlo, and members of the Kayser and Sehgal Labs for helpful guidance and input. We thank Jonathan Levenson and Neil Singh for technical support. We thank E Kravitz and members of the Kravitz lab (in particular B Chowdhury) for introducing MSK to the field of fly aggression, and K Asahina for generous technical guidance. This work was funded by NIH grants K08 NS090461 and T32 HL07713 (MSK) and the Howard Hughes Medical Institute (AS).

## Additional information

### Funding

| Funder | Grant reference | Author |
| --- | --- | --- |
| National Institute of Neurological Disorders and Stroke (NINDS) | K08 NS090461 | Matthew S Kayser |
| National Heart, Lung, and Blood Institute (NHBLI) | T32 HL07713 | Matthew S Kayser |

| Funder | Grant reference | Author |
| --- | --- | --- |
| Howard Hughes Medical Institute (HHMI) | | Amita Sehgal |

The funders had no role in study design, data collection and interpretation, or the decision to submit the work for publication.

## Author contributions

MSK, Conception and design, Acquisition of data, Analysis and interpretation of data, Drafting or revising the article; BM, Acquisition of data, Analysis and interpretation of data; ZY, Acquisition of data, Contributed unpublished essential data or reagents; AS, Conception and design, Drafting or revising the article

## Author ORCIDs

Amita Sehgal, http://orcid.org/0000-0001-7149-8588

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
