## [Decision Letter]

Thank you for submitting your work entitled “Sleep deprivation suppresses aggression in *Drosophila*” for peer review at *eLife*. Your submission has been favorably evaluated by K VijayRaghavan (Senior editor), Leslie Griffith, a Reviewing editor, and two additional reviewers.

The reviewers have discussed the reviews with one another and the Reviewing editor has drafted this decision to help you prepare a revised submission.

Summary:

This paper looks at the linkage between sleep deprivation and aggression. The mammalian literature has both human studies and rodent studies, but no clear consensus for the relationship has emerged – different studies report increased, unchanged or decreased aggression. Potentially the roles of stress and context may explain some of this variability, but examining the question in flies where there are relatively robust and standardized conditions to assess both is an opportunity to shed light on how sleep deprivation might affect aggression and the mechanisms that underlie the linkage. This paper provides a clear result suggesting that sleep deprivation decreases aggression without disrupting other complex behaviors.

Essential revisions:

The authors want to suggest that social amelioration of aggression and sleep deprivation's anti-aggression effects are mechanistically different based on the ability of an OA agonist, but not a DA agonist, to rescue *dTrpA1* SD from either *Tdc2* or TH driver. All the reviewers were concerned that thermogenetic sleep deprivation might represent a special case of sleep deprivation since the brain is flooded with bioaminergic NT for 12h and there can be lasting effects. If there is truly a distinction between the roles of OA and DA in aggression after sleep deprivation, the pharmacology should also hold for high temp sleep deprivation and mechanical sleep dep. Without those data it is really not appropriate to make the general statement that “sleep deprivation” is different than shared housing – you need to do all types of deprivation to make this claim, which is a central point of the paper.

Minor points:

1) Why are there regular 2h oscillations in all the sleep data? To my knowledge there is no ultradian rhythm to sleep and this suggests strongly to me that there was some external disturbance of sleep in all these experiments. Were they all done the same incubator? Are light or temperature levels oscillating with the same rhythm? The data shown in this paper are clearly not taken in constant conditions and while this may not affect the outcome of the experiments it is not completely clear to me that it wouldn't. Retaking one or two key data sets in conditions that are truly constant would allay this concern.

2) Can the authors clarify how they distinguish treatment groups in assays where different types of males are compared to one another. The text associated with Figure 4 indicates that they used painted dots. Is this also true for the data shown in Figure 1? I couldn't tell when I looked at the movies (first movie the males had dots, second movie did not appear to have dots?). Please clarify.

3) No control for *TH-GAL4*? Only *Tdc2-GAL4* and *UAS-dTrpA1* are shown.

---

## [Author Response]

*The authors want to suggest that social amelioration of aggression and sleep deprivation's anti-aggression effects are mechanistically different based on the ability of an OA agonist, but not a DA agonist, to rescue d*TrpA1 *SD from either* Tdc2 *or TH driver. All the reviewers were concerned that thermogenetic sleep deprivation might represent a special case of sleep deprivation since the brain is flooded with bioaminergic NT for 12h and there can be lasting effects. If there is truly a distinction between the roles of OA and DA in aggression after sleep deprivation, the pharmacology should also hold for high temp sleep deprivation and mechanical sleep dep. Without those data it is really not appropriate to make the general statement that* “*sleep deprivation*” *is different than shared housing – you need to do all types of deprivation to make this claim, which is a central point of the paper*.

To determine whether “there is truly a distinction between the roles of OA and DA in aggression after sleep deprivation”, we focused on pharmacologic rescue following mechanical sleep deprivation because it is the most robust with regard to effect on aggression. Consistent with results following thermogenetic sleep deprivation, we find that CDM but not L-DOPA rescues fighting after mechanical sleep deprivation. This experiment confirms that selectivity of the OA agonist to rescue aggression is not specific to a single mode of sleep loss.

To provide further evidence that thermogenetic and mechanical sleep deprivation engage similar mechanisms, we now also show that thermogenetic sleep deprivation (like mechanical) results in impaired performance in a competitive copulation assay the following day. Moreover, we demonstrate that the OA agonist but not DA agonist rescues sexual fitness following sleep deprivation, consistent with rescue of aggression (see Figure 4).

*Minor points*:

*1) Why are there regular 2h oscillations in all the sleep data? To my knowledge there is no ultradian rhythm to sleep and this suggests strongly to me that there was some external disturbance of sleep in all these experiments. Were they all done the same incubator? Are light or temperature levels oscillating with the same rhythm? The data shown in this paper are clearly not taken in constant conditions and while this may not affect the outcome of the experiments it is not completely clear to me that it wouldn't. Retaking one or two key data sets in conditions that are truly constant would allay this concern*.

We thank the reviewers for pointing this out. The sleep plots are noisy due to the small number of flies used – the noise looks most prominent in Figure 3 and Figure 2—figure supplement 2. In both instances, a single experiment with a small number of flies was used to generate the sleep plot; not all flies from each independent experiment that generates the lunging data are represented in these sleep traces. Both of those particular sleep plots come from experiments run at nearly the same time. While we agree it looks as though there are regular oscillations, we have confirmed by environmental monitors that temperature, humidity, and light are constant. It is possible that incubator fan noise or vibration contributed to that pattern. However, we now present new sleep plots from experiments run at a different time that make clear there are no ultradian rhythms in our experiments. These sleep experiments are from flies that yield the same effect on aggression as the noisier plots previously displayed.

*2) Can the authors clarify how they distinguish treatment groups in assays where different types of males are compared to one another. The text associated with*
Figure 4
*indicates that they used painted dots. Is this also true for the data shown in*
Figure 1*? I couldn't tell when I looked at the movies (first movie the males had dots, second movie did not appear to have dots?). Please clarify*.

We thank the reviewers for bringing attention to this oversight on our part. We did paint flies in all “between condition” experiments in Figure 1, as well as Figure 4. We have now updated the text and methods to reflect this approach.

*3) No control for* TH-GAL4*? Only* Tdc2-GAL4 *and* UAS-dTrpA1 *are shown*.

We have added experiments showing that *TH-GAL4* controls do not show a change in lunging following exposure to 29C overnight, while sleep deprivation with *TH-GAL4> UAS-dTrpA1* causes reduced aggression.